

# Characterization of the mitochondrial genome of *Arge bella* Wei & Du sp. nov. (Hymenoptera: Argidae)

Shiyu Du[1,*], Gengyun Niu[2,*], Tommi Nyman[3] and Meicai Wei[1]

[1] Central South University of Forestry and Technology, Key Laboratory of Cultivation and Protection for Non-Wood Forest Trees (Central South University of Forestry and Technology), Ministry of Education, Changsha, Hunan, China

[2] Jiangxi Normal University, Life Science College, Nanchang, Jiangxi, China

[3] Norwegian Institute of Bioeconomy Research, Department of Ecosystems in the Barents Region, Svanhovd Research Station, Svanvik, Norway

[*] These authors contributed equally to this work.

## ABSTRACT

We describe *Arge bella* Wei & Du sp. nov., a large and beautiful species of Argidae from south China, and report its mitochondrial genome based on high-throughput sequencing data. We present the gene order, nucleotide composition of protein-coding genes (PCGs), and the secondary structures of RNA genes. The nearly complete mitochondrial genome of *A. bella* has a length of 15,576 bp and a typical set of 37 genes (22 tRNAs, 13 PCGs, and 2 rRNAs). Three tRNAs are rearranged in the *A. bella* mitochondrial genome as compared to the ancestral type in insects: *trnM* and *trnQ* are shuffled, while *trnW* is translocated from the *trnW*-*trnC*-*trnY* cluster to a location downstream of *trnI*. All PCGs are initiated by ATN codons, and terminated with TAA, TA or T as stop codons. All tRNAs have a typical cloverleaf secondary structure, except for *trnS1*. H821 of *rrnS* and H976 of *rrnL* are redundant. A phylogenetic analysis based on mitochondrial genome sequences of *A. bella*, 21 other symphytan species, two apocritan representatives, and four outgroup taxa supports the placement of Argidae as sister to the Pergidae within the symphytan superfamily Tenthredinoidea.

## INTRODUCTION

With about 950 valid species in the world, Argidae is the second-largest family of the paraphyletic suborder Symphyta of the order Hymenoptera (*Choi et al., 2016*). Eastern Asia is one of the three main centers of diversity of the family (*Wei & Nie, 1997*). Within China, about 170 species and 16 genera have been recorded (*Choi et al., 2016*). However, there are probably many more species to be described or newly recorded from this vast country.

Symphyta is the predominantly herbivorous and relatively less diverse suborder of the Hymenoptera, and contains more than 8,500 described species (*Taeger, Blank & Liston, 2010*). The systematic arrangement of Symphyta, including the numbers of families and

Corresponding author
Meicai Wei, weimc@126.com, weim@jxnu.edu.cn

superfamilies are quite uncertain. Most symphytan researchers have divided the extant Symphyta into 14 families (*Benson, 1938*; *Königsmann, 1977*; *Abe & Smith, 1991*; *Taeger, Blank & Liston, 2010*) under four (*Benson, 1938*), six (*Königsmann, 1977*; *Abe & Smith, 1991*) or seven superfamilies (*Taeger, Blank & Liston, 2010*), and the six superfamilies in the system of Königsmann and of Abe and Smith are also different. *Rasnitsyn (1988)* divided extant taxa of Symphyta into two suborders, five infraorders, and 13 families. *Wei & Nie (1997)*; *Wei & Nie (1998)* divided the original Symphyta into five suborders, eleven superfamilies, and 20 families. It seems that a consensus on the systematics of Symphyta among sawfly researchers is difficult to get just based on morphological analysis, so using molecular-genetic data would be critical to test the different proposed systems and to approach a natural system of Symphyta.

The monophyly of Tenthredinoidea is supported by both morphological (*Wei & Nie, 1997*) and molecular data (*Malm & Nyman, 2015*) as well as combined analyses (*Ronquist et al., 2012a*; *Sharkey et al., 2012*; *Klopfstein et al., 2013*), but relationships among core tenthredinoids are less clear. Argidae was inferred as the sister to the remaining tenthredinoids by *Wei & Nie (1997)*, but the disaccord of this result with several recent studies may be arisen from the limited dataset. *Malm & Nyman (2015)* analyzed nine protein-coding genes of 164 taxa to reconstruct the phylogenetic backbone of the Hymenoptera. In their analysis, 13 taxa were included to represent five out of seven subfamilies within Argidae; in the tree, Argidae and Pergidae form a monophylum as sister to the other non-blasticotomid tenthredinoids, which supports more comprehensive morphological (*Vilhelmsen, 1997*; *Schmidt et al., 2006*) and combined studies (*Ronquist et al., 2012a*; *Sharkey et al., 2012*; *Klopfstein et al., 2013*). A recent analysis of whole-body transcriptomes also inferred the monophylum of Argidae and Pergidae (*Peters et al., 2017*).

The mitochondrial genomes of 21 symphytan species have been reported (Table 1; data were collected at NCBI, available at https://www.ncbi.nlm.nih.gov/; accessed 3 Nov. 2017). Five phylogenetic analyses have been conducted based on nucleotide sequences of symphytan mitochondrial genomes (*Castro & Dowton, 2005*; *Dowton et al., 2009b*; *Song et al., 2015b*; *Song et al., 2016*; *Doğan & Korkmaz, 2017*), but none of them have provided clear insights into symphytan relationships because of the taxonomically restricted respresentation of sawfly families in the datasets: the mitochondrial genomes of Argidae, Xyelidae, Diprionidae, Heptamelidae, Blasticotomidae, Megalodontesidae, Pamphiliidae, Xiphydriidae, Siricidae, and Anaxyelidae have not been previously reported.

The small number of available sawfly mitochondrial genomes also limits our understanding of their genomic architecture. Compared with the ancestral gene arrangement of insects, only translocated and swapped are exhibited in *A. bella*. The conservation of rRNA secondary structures exceeds that of its nucleotides and, therefore, it is recommended that secondary structures guide decisions about the alignment of homologous positions for phylogenetic studies (*Kjer, 1995*). The secondary structures of *rrnS* of *A. bella* and *Cephus* species are conservative in H821, but previous researchers supported that Argidae and Pergidae form a monophylum as sister to the remaining tenthredinoids (*Malm & Nyman, 2015*), instead of Argidae formed a sister group with *Cephus* species. However, inferred secondary structures can only be considered as working

**Table 1  General information of the mitochondrial genomes of Symphyta.**

| Species | Length (bp) | Completeness | Family | Subfamily | Accession number | Resources |
|---------|-------------|--------------|--------|-----------|------------------|-----------|
| *Perga condei* | 13,416 bp | partial | Pergidae | Perginae | AY787816 | *Castro & Dowton (2005)* |
| *Orussus occidentalis* | 15,947 bp | complete | Orussidae | Orussinae | FJ478174 | *Dowton et al. (2009a)* |
| *Trichiosoma anthracinum* | 15,392 bp | partial | Cimbicidae | Cimbicinae | KT921411 | *Song et al. (2016)* |
| *Corynis lateralis* | 14,999 bp | partial | Cimbicidae | Coryninae | KY063728 | *Doğan & Korkmaz (2017)* |
| *Monocellicampa pruni* | 15,169 bp | partial | Tenthredinidae | Hoplocampinae | JX566509 | *Wei, Wu & Liu (2015)* |
| *Allantus luctifer* | 15,418 bp | complete | Tenthredinidae | Allantinae | KJ713152 | *Wei, Niu & Du (2014)* |
| *Asiemphytus rufocephalus* | 14,864 bp | partial | Tenthredinidae | Allantinae | KR703582 | *Song et al. (2016)* |
| *Tenthredo tienmushana* | 14,942 bp | partial | Tenthredinidae | Tenthredininae | KR703581 | *Song et al. (2015a)* and *Song et al. (2015b)* |
| *Cephus cinctus* | 19,339 bp | complete | Cephidae | Cephinae | FJ478173 | *Dowton et al. (2009a)* |
| *Cephus pygmeus* | 16,145 bp | partial | Cephidae | Cephinae | KM377623 | *Korkmaz et al. (2015)* |
| *Cephus sareptanus* | 15,212 bp | partial | Cephidae | Cephinae | KM377624 | *Korkmaz et al. (2015)* |
| *Calameuta filiformis* | 20,055 bp | complete | Cephidae | Cephinae | KT260167 | *Korkmaz et al. (2016)* |
| *Calameuta idolon* | 19,746 bp | complete | Cephidae | Cephinae | KT260168 | *Korkmaz et al. (2016)* |
| *Trachelus iudaicus* | 20,370 bp | complete | Cephidae | Cephinae | KX257357 | *Korkmaz et al. (2017)* |
| *Trachelus tabidus* | 18,539 bp | complete | Cephidae | Cephinae | KX257358 | *Korkmaz et al. (2017)* |
| *Hartigia linearis* | 20,116 bp | partial | Cephidae | Hartigiinae | KX907843 | *Korkmaz et al. (2018)* |
| *Janus compressus* | 16,700 bp | partial | Cephidae | Hartigiinae | KX907844 | *Korkmaz et al. (2018)* |
| *Pachycephus cruentatus* | 14,568 bp | partial | Cephidae | Hartigiinae | KX907845 | *Korkmaz et al. (2018)* |
| *Pachycephus smyrnensis* | 15,203 bp | partial | Cephidae | Hartigiinae | KX907846 | *Korkmaz et al. (2018)* |
| *Syrista parreyssii* | 15,924 bp | partial | Cephidae | Hartigiinae | KX907847 | *Korkmaz et al. (2018)* |
| *Characopygus scythicus* | 10,558 bp | partial | Cephidae | Hartigiinae | KX907848 | *Korkmaz et al. (2018)* |

hypotheses, and would be almost impossible to estimate without using a comparative approach (*Misof & Fleck, 2003*). Our understanding of the secondary structures of symphytan rRNAs has been developed only from seven Cephidae species (*Dowton et al., 2009a*; *Korkmaz et al., 2015*; *Korkmaz et al., 2016*; *Korkmaz et al., 2017*). More representatives and comparative analyses are therefore required within the Symphyta.

Here, we describe a large and beautiful new species of *Arge Schrank (1802)* and report its near-complete mitochondrial genome sequence, as the first representative of the family Argidae. We characterize the nucleotide composition, codon usage and secondary structure of tRNAs of this mitochondrial genome. We compare the gene rearrangement of *A. bella* with the ancestral gene arrangement of insects. We also analyze two rRNAs secondary structures across the sequenced symphytan mitochondrial genomes. The structural differences of rRNAs between *A. bella* and *Cephus* species are described to establish structural features as potentially useful characters for symphytan systematics. Finally, we report the results of phylogenetic analyses that we used to verify the phylogenetic placement of *A. bella* based on sequences of 13 protein-coding genes and two rRNA genes of 22 species of Symphyta, two representatives of Apocrita, and four outgroup taxa. Our results support the placement of Argidae as sister to the Pergidae within the symphytan superfamily Tenthredinoidea.

## MATERIALS AND METHODS

### Description of new species

Specimens were examined with a Leica S8APO dissection microscope. Adult images were taken with a Nikon D700 digital camera, and sequentially focused images were montaged using Helicon Focus (HeliconSoft), while detailed images were taken with Leica Z16 APO/DFC550. All images were further processed with Adobe Photoshop CS 6.0.

The terminology of sawfly genitalia follows *Ross (1945)*, and that of general morphology follows *Viitasaari (2002)*. Abbreviations used are: OOL = distance between the eye and outer edge of lateral ocellus; POL = distance between the mesal edges of the lateral ocelli; OCL = distance between a lateral ocellus and the occipital carina or hind margin of the head.

The holotype and all paratypes of the new species are deposited in the Insect Collection of Central South University of Forestry and Technology, Changsha, Hunan, China (CSCS).

All nomenclatural acts, authors and literature are registered in Zoobank as per the recent proposed amendment to the International Code of Zoological nomenclature for a universal register for animal names (*Polaszek et al., 2005a*; *Polaszek et al., 2005b*; *Pyle, Earle & Greene, 2008*; *ICZN, 2008*). Rules for spelling Chinese personal and place names follow GB/T 16159-1996 and ISO 7098: 1991: Chinese people's names are to be written separately with the surname first, followed by the personal name written as one word, with the initial letters of both capitalized.. Chinese place names should be alphabetized according to the Spelling Rules for Chinese Geographical Place Names, document no. 17 (1984) of the State Committee on Chinese Geographical Place Names. Separate the geographical proper name from the geographical feature name and capitalize the first letter of both (*Niu, Wei & Taeger, 2012*).

The electronic version of this article in Portable Document Format (PDF) will represent a published work according to the International Commission on Zoological Nomenclature (ICZN), and hence the new names contained in the electronic version are effectively published under that Code from the electronic edition alone. This published work and the nomenclatural acts it contains have been registered in ZooBank, the online registration system for the ICZN. The ZooBank LSIDs (Life Science Identifiers) can be resolved and the associated information viewed through any standard web browser by appending the LSID to the prefix http://zoobank.org/. The LSID for this publication is: urn:lsid:zoobank.org:pub:A94BD62A-D4BE-40F9-8718-84F425875C7C. The online version of this work is archived and available from the following digital repositories: PeerJ, PubMed Central and CLOCKSS.

### Library construction and sequencing

Total genomic DNA of a single specimen was used for library preparation with insert size of 250 bp following the manufacturer's instructions, and then 150 bp PE sequenced on an Illumina HiSeq 4,000 platform for around 2.5 Gb of data at BGI-Shenzhen, China. The sequencing reads have been deposited in NCBI SRA database under accession number: PRJNA493965.

## Mitochondrial genome assembly

Pre-analysis data filtering included: (i) Clean data was generated following published protocols (*Zhou et al., 2013*; *Tang et al., 2014*; *Tang et al., 2015*), by removing reads with adaptor contamination, >10% low-quality bases, or >5 bp Ns; (ii) clean data was then compared with reference mitochondrial genomes downloaded from GenBank (716 RefSeq genomes, including 699 arthropods, seven starfish and 10 cyprinid fish; accessed on 10 March 2014) to screen out candidate mitochondrial reads using relaxed criteria: BLAST identity >30% and *E*-value <0.00001; (iii) a 51-mer set was then generated from these candidate mito-reads and used as reference for a second round of data filtering for the discarded reads from step 2; (iv) *De novo* assembly was performed using *SOAPdenovo-Trans* (*Xie et al., 2014*) (-K 71, -L 100, -t 1), *SOAPdenovo* 2.0 (*Luo et al., 2012*; *Li et al., 2010*) (-K 61, -k 45), *IDBA-UD* (*Peng et al., 2012*) (kMaxShortSequence = 256, –num threads 12), and mitochondrial protein-encoding assemblies and mitochondrial genome-sequence candidates were annotated by a custom *Perl* script (*Zhou et al., 2013*) using RefSeq mitochondrial genomes of target animal taxa (604 arthropod species, two asteriid starfish and the zebrafish; downloaded from GenBank on 13 June 2013) downloaded from NCBI as reference. The mitochondrial genome was constructed, corrected and manually checked as previously described (*Tang et al., 2015*).

## Mitochondrial genome annotation and secondary structure prediction

All of the tRNAs were identified using MITOS (http://mitos.bioinf.uni-leipzig.de/index.py) (*Bernt et al., 2013*) using the default settings. The initiation and termination codons of PCGs were determined in Geneious v8.0.5 (*Kearse et al., 2012*) (available from http://www.geneious.com) using reference sequences from other symphytan species, and then checked manually. Secondary structures of rRNAs were inferred using alignment to the models predicted for *Trichiosoma anthracinum* and *Labriocimbex sinicus* (Yan et al., in press). The predicted secondary structures of tRNAs and rRNAs were drawn using VARNA v3-93 (*Darty, Denise & Ponty, 2009*) and RNAviz 2.0.3 (*De Rijk, Wuyts & Wachter, 2003*). Helix numbering follows the convention established at the CRW site (*Cannone et al., 2002*) and *Apis mellifera* rRNA secondary structure (*Gillespie et al., 2006*) with minor modifications.

The A + T content of nucleotide sequences and relative synonymous codon usage (RSCU) were calculated using MEGA v7.0 (*Kumar, Stecher & Tamura, 2016*). Strand asymmetry was calculated using the formulae by *Perna & Kocher (1995)*: $GC-skew = (G-C)/(G+C)$ and $AT-skew = (A-T)/(A+T)$.

## Phylogenetic analyses

Phylogenetic analyses were performed based on aligned sequences of the 13 PCGs and two rRNAs of the nearly complete mitochondrial genome of *A. bella* and 21 other symphytan mitochondrial genomes downloaded from GenBank (Table 1). These additional taxa represented five families: Tenthredinidae (*Wei, Wu & Liu, 2015*; *Wei, Niu & Du, 2014*; *Song et al., 2015a*; *Song et al., 2016*), Cimbicidae (*Song et al., 2016*; *Doğan & Korkmaz, 2017*), Pergidae (*Castro & Dowton, 2005*), Orussidae (*Dowton et al., 2009a*), and Cephidae

(*Dowton et al., 2009a*; *Korkmaz et al., 2015*; *Korkmaz et al., 2016*; *Korkmaz et al., 2017*; *Korkmaz et al., 2018*). As the Symphyta is paraphyletic with respect to the suborder Apocrita, we also included the mitochondrial genomes of the apocritan species *Parapolybia crocea* (GenBank: KY679828) and *Taeniogonalos taihorina* (GenBank: NC027830) in the analysis. As outgroups, we included *Neopanorpa pulchra* (GenBank: FJ169955) from Mecoptera, *Anopheles gambiae* (GenBank: L20934) from Diptera, *Neochauliodes parasparsus* (GenBank: KX821680) from Megaloptera and *Paroster microsturtensis* (GenBank: MG912997) from Coleoptera.

Nucleotide sequences of the 13 PCGs from the mitochondrial genomes of *A. bella* and the 27 other included species were translated into amino acid sequences and then aligned by MUSCLE in MEGA v7.0. Nucleotide sequences of two rRNAs from the included taxa were aligned by MAFFT (https://www.ebi.ac.uk/Tools/msa/mafft/). The amino acid alignments of the 13 PCGs and two rRNAs were concatenated using SequenceMatrix v1.7.8 (*Vaidya, Lohman & Meier, 2011*) and used in phylogenetic analyses under the Maximum-likelihood (ML) criteron and Bayesian inference (BI). Partition schemes and substitution models were calculated simultaneously in PartitionFinder v1.1.1 (*Lanfear et al., 2012*). The branch lengths and search strategy of schemes were set as linked and greedy, and models were selected based on AICc and BIC. The GTR+I+G model was chosen as the best-fitting model for all partitions for both ML and BI analyses.

The ML analysis was performed using the IQ-TREE web server (http://iqtree.cibiv.univie.ac.at/) (*Trifinopoulos et al., 2016*), using default parameters except for 0.1 as the perturbation strength and 1000 as the IQ-TREE stopping rule. The BI analysis was performed using MrBayes v3.2.6 (*Ronquist et al., 2012b*) on the CIPRES Science Gateway (*Miller, Pfeiffer & Schwartz, 2010*). Rate and substitution-model parameters were unlinked across partitions. Two independent runs with four simultaneous Markov chains (one cold, three incrementally heated at $T = 0.2$) were run for five million generations, with sampling of parameters and trees occurring every 1,000 generations. The maximum clade credibility tree showing all compatible groupings was calculated with a burn-in fraction of 10%, after confirming in Tracer v1.6.0 (*Rambaut & Drummond, 2013*) (Available at: http://beast.bio.ed.ac.uk/Tracer) that both runs had converged and that appropriate effective sample sizes were achieved for sampled parameters. Trees were edited in FigTree v1.4.2 (http://tree.bio.ed.ac.uk/software/figtree/).

# RESULTS AND DISCUSSION

### *Arge bella* Wei & Du sp. nov.

urn:lsid:zoobank.org:act:9AF00C3F-D5EE-474B-BD9E-4794BECACA4F

**Etymology.** This species is named after its beautiful body colour.

**Holotype.** Female. China: Hunan Province, Guidong Conty, Mt. Qiyun, Hydropower Station Valley, alt. 752m, 25°45.361′N, 113°55.598′E, April 4, 2015, Yuchen Yan, Ting Liu leg. (CSCS).

**Paratypes.** 2 Females. Locality and collecting time as the holotype, Hang Zhao, Mengmeng Liu leg. (CSCS).

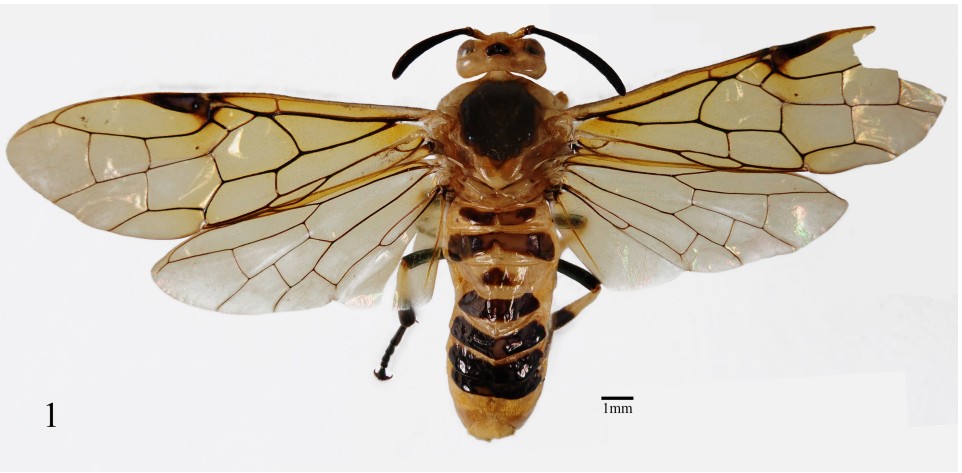

**Figure 1** *Arge bella* **Wei & Du sp. nov. Adult female, dorsal view.** Scale bars = 1 mm.

**Distribution.** China (Hunan).

**Remarks.** This new species is somewhat similar to *Arge nigricrux Malaise (1943)* and *A. vitalisi Turner (1919)*, but differs from these two species by the followings: the antennal flagellum entirely black; dorsum of mesonotum blue black except for posterior of mesoscutellum; abdomen yellow brown, terga 1–2 and 4–7 largely blue black; wings distinctly yellowish, without transverse macula; hind tarsus entirely blue black; mesepisternum blue black with upper fourth yellow brown. In *A. nigricrux* and *A. vitalisi*, the antennal flagellum and mesonotum in female entirely yellow; abdomen yellow brown, terga 4 largely black, terga 5–8 with short middle black maculae; wings weakly yellowish, with distinct transverse smoky macula just below pterostigma; hind tarsus entirely yellow brown; mesepisternum entirely black or entirely yellow brown.

## Description

**Female.** Body length 13 mm (Fig. 1). Body and leg yellow brown; apex of mandible, apical half of pedicel and flagellum entirely black, without bluish tinge; ocellar area, entire mesoscutal middle lobe, dorsum of mesoscutal lateral lobe, anterior triangular lobe of mesoscutellum, mesepisternum except for dorsal fourth, katepimeron except for margins, small central macula on metapleuron, large transversal band on tergum 1, dorsum of terga 2 and 4–7, small middle macula on tergum 3, apical 0.8 of middle and hind femora, apex of middle tibia, apex of middle basitarsus and tarsomeres 2–5, apical 0.4 of hind tibia and entire hind tarsus black with distinct bluish tinge, upper margin of black macula on mesepisternum convex (Fig. 2C); most of body hairs yellow brown, hairs on flagellum, middle and hind femora, spines on inner sides of sheath mostly black, hairs on mesonotum dark brown. Wings hyaline, with distinct yellowish tinge, veins C, r1 and A largely pale brown, pterostigma and other veins black brown.

Head smooth, shiny; basal half of mandible with large and dense punctures; clypeus, face, lower half of inner orbit and frontal wall distinctly and rather densely punctured,

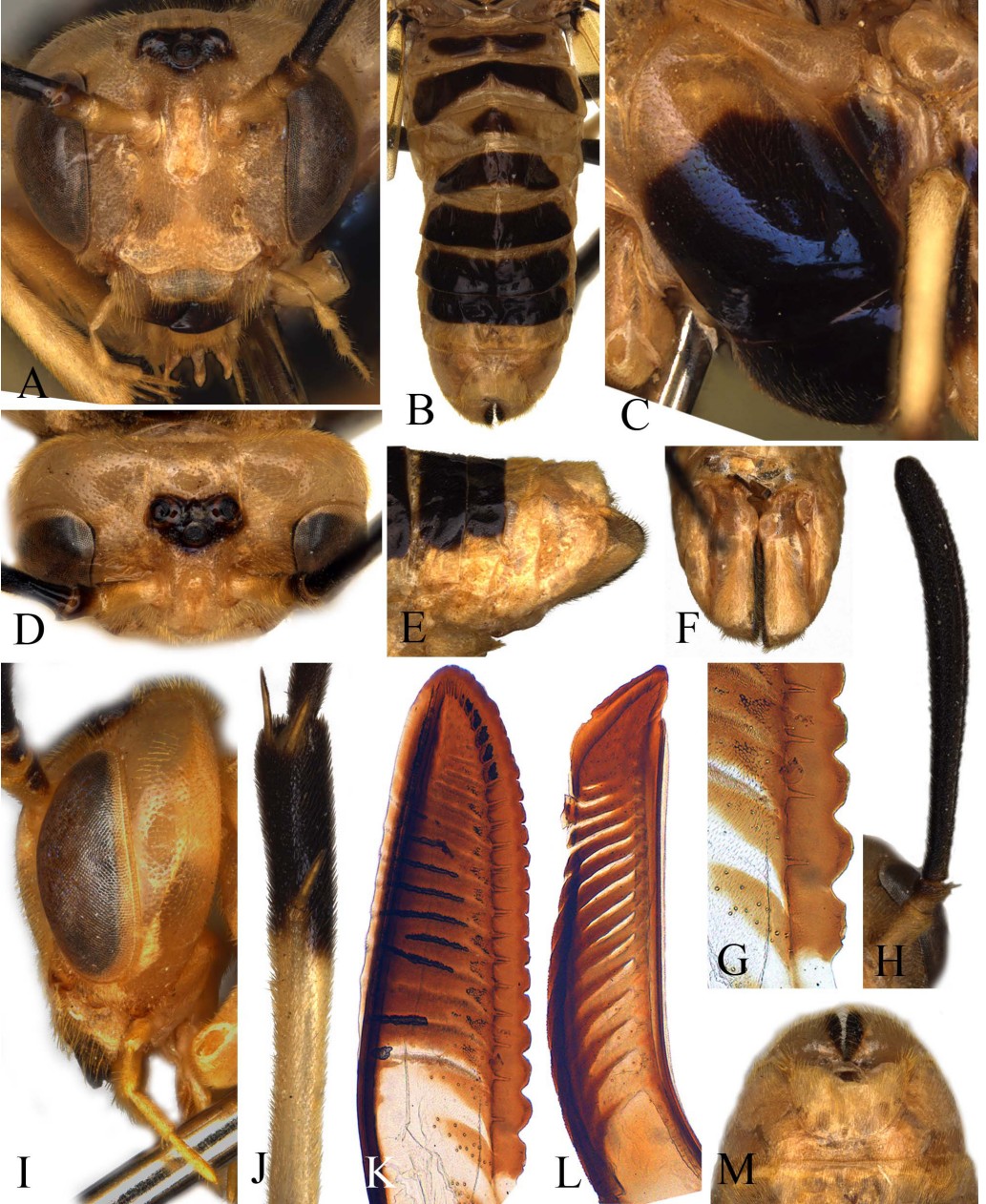

**Figure 2** *Arge bella.* Figures (A-M). *Arge bella* Wei & Du sp. nov. (A) Head of female, front view; (B) abdomen, dorsal view; (C) mesopleuron of female, lateral view; (D) head of female, dorsal view; (E) sheath of female, lateral view; (F) sheath of female, ventral view; (G) basal serrulae of lancet; (H) Antenna of female; (I) head of female, lateral view; (J) tibia of hind leg; (K) lancet; (L) lance; (M) sheath of female, dorsal view.

temple densely and minutely punctured, postocellar area hardly punctured, postorbit minutely and sparsely punctured, frontal basin smooth; pronotum finely and faintly punctured; mesonotum minutely and sparsely punctured, most of mesoscutellum smooth with some minute punctures, parapsis smooth, impunctate and without microsculptures;

metanotum and mesepisternum smooth, hardly punctured; anepimeron minutely punctured, katepimeron finely microsculptured; center of mesosternum distinctly punctured; metapleuron impunctate, without microsculptures; dorsum of abdominal terga smooth, impunctate, terga 1–2 faintly microsculptured, sterna and sheath smooth.

Labrum about 2 times as broad as long, apex obtusely truncate; clypeus flat, anterior incision shallow and roundish; supraclypeal area roundly elevated, without middle ridge; lateral carinae between antennal toruli low and obtuse, almost parallel downwards, not merged together, largest breadth between lateral carinae about 1.5 times diameter of median ocellus (Fig. 2A); middle fovea round, distinct, shallowly open to frontal basin; frons small, center evenly concave, frontal wall distinct; malar space 0.6 times diameter of median ocellus; inner margin of eyes parallel, distance between eyes 1.1 times longest axis of eye; POL: OOL: OCL = 20: 27: 22; postocellar area flat, breadth 1.7 times length; lateral furrows shallow and fine, slightly convergent backwards; postocellar furrow fine and shallow, interocellar furrow broad and shallow; in dorsal view (Fig. 2D), head about 0.65 times as long as eye, hardly enlarged behind eyes; head in lateral view as in Fig. 2I. Antenna slightly enlarged toward apex, hardly bent, longitudinal carina low and faint, pedicel as long as broad, flagellum 0.8 times as long as thorax and about 1.35 times head breadth (Fig. 2I). Middle furrow on mesoscutal middle lobe faint, notauli distinct; mesoscutellum 1.2 times as broad as long, anterior 0.2 with a short middle furrow; distance between cenchri 0.2 times breadth of a cenchrus. Forewing: vein R short, about 0.7 times as long as vein and 0.6 times as long as free abscissa of vein Sc, vein R+M about half length of vein 1r-m, second abscissa of vein Rs clearly longer than third abscissa of Rs, third abscissa of Rs 1.4 times as long as fourth abscissa of Rs, cell 1Rs clearly longer than cell 2Rs, upper and lower margins of cell 2Rs equal in length; vein 2Rs roundly convex outwards, vein cu-a meeting cell 1M at basal 0.4, basal anal cell closed. Hind wing: cell Rs 1.25 times as long as cell M, cell M about 2.1 times as long as broad; anal cell closed; relative length of cells A, petiole of anal cell and cu-a about 80: 57: 27; outer margin of fore and hind wings naked (Fig. 1). Middle and hind tibiae each with 1 preapical spur (Fig. 2J); hind basitarsus slightly longer than following 3 tarsomeres together. Ovipositor sheath as long as hind femur, basal third distinctly concave in lateral view (Fig. 2F); apex of sheath round in dorsal view (Fig. 2M); subapical part of lance weakly narrowed (Fig. 2L); lancet broad, annular spines very short, serrulae strongly protruding (Fig. 2G), basal and middle serrulae as in Fig. 2G.
**Male.** Unknown.

## Architecture and nucleotide composition of *A. bella* mitochondrial genome

We sequenced the nearly complete mitochondrial genome of *A. bella*, which deposited in GenBank of NCBI under the accession number MF287761. The sequenced region is 15,576 bp in length, with 13 protein-coding, two rRNA genes and 22 tRNA genes. Of these, 23 genes (9 PCGs and 14 tRNAs) were encoded by the J strand, while the remaining ones were encoded by the N strand (Table 2).

Compared with the ancestral gene arrangement of insects, the mitochondrial genome of *A. bella* exhibited only few rearrangements: *trnM* and *trnQ* have swapped positions,

**Table 2  Mitochondrial genome characteristics of *A bella*.**

| Gene | Strand | Start | Stop | Length(bp) | Start codon | Stop codon | Codon | IGN |
|------|--------|-------|------|-----------|-------------|------------|-------|-----|
| trnW | J | 152 | 219 | 68 | | | UGA | |
| trnI | J | 253 | 323 | 71 | | | AUC | 33 |
| trnM | J | 354 | 422 | 69 | | | AUG | 30 |
| trnQ | N | 420 | 488 | 69 | | | CAA | −3 |
| ND2 | J | 486 | 1,562 | 1,077 | ATA | TAA | | −3 |
| trnC | N | 1,575 | 1,643 | 69 | | | UGC | 12 |
| trnY | N | 1,651 | 1,722 | 72 | | | UAC | 7 |
| COX1 | J | 1,723 | 3,271 | 1,549 | ATT | T | | 0 |
| trnL2 | J | 3,272 | 3,336 | 65 | | | UUA | 0 |
| COX2 | J | 3,337 | 4,017 | 681 | ATG | TAA | | 0 |
| trnK | J | 4,062 | 4,133 | 72 | | | AAG | 44 |
| trnD | J | 4,135 | 4,205 | 71 | | | GAC | 1 |
| ATP8 | J | 4,207 | 4,380 | 120 | ATA | TAA | | 1 |
| ATP6 | J | 4,374 | 5,048 | 675 | ATG | TAA | | −7 |
| COX3 | J | 5,048 | 5,828 | 781 | ATG | T | | −1 |
| trnG | J | 5,829 | 5,894 | 66 | | | GGA | 0 |
| ND3 | J | 5,895 | 6,248 | 354 | ATA | TAA | | 0 |
| trnA | J | 6,264 | 6,328 | 65 | | | GCA | 15 |
| trnR | J | 6,329 | 6,393 | 65 | | | CGA | 0 |
| trnN | J | 6,419 | 6,491 | 73 | | | AAC | 25 |
| trnS1 | J | 6,492 | 6,554 | 63 | | | AGC | 0 |
| trnE | J | 6,556 | 6,622 | 67 | | | GAA | 1 |
| trnF | N | 6,621 | 6,688 | 68 | | | UUC | −2 |
| ND5 | N | 6,694 | 8,409 | 1,716 | ATT | TAA | | 5 |
| trnH | N | 8,410 | 8,482 | 73 | | | CAC | 0 |
| ND4 | N | 8,483 | 9,821 | 1,339 | ATG | T | | 0 |
| ND4L | N | 9,815 | 10,111 | 297 | ATT | TAA | | −7 |
| trnT | J | 10,114 | 10,179 | 66 | | | ACA | 2 |
| trnP | N | 10,180 | 10,246 | 67 | | | CCA | 0 |
| ND6 | J | 10,248 | 10,777 | 530 | ATA | TA | | 1 |
| CYTB | J | 10,778 | 11,911 | 1,134 | ATG | TAA | | 0 |
| trnS2 | J | 11,917 | 11,985 | 69 | | | UCA | 5 |
| ND1 | N | 11,996 | 12,949 | 954 | ATT | TAA | | 10 |
| trnL1 | N | 12,950 | 13,018 | 69 | | | CUA | 0 |
| rrnL | N | 13,019 | 14,408 | 1,390 | | | | 0 |
| trnV | N | 14,409 | 14,475 | 67 | | | GUA | 0 |
| rrnS | N | 14,476 | 15,314 | 839 | | | | 0 |

**Notes.**
J and N refers to heavy and light strands, respectively; IGN refer to intergenic nucleotides. Minus indicates overlapping sequences between adjacent genes.

and *trnW* has been translocated from the *trnW-trnC-trnY* cluster to downstream of *trnI* (Fig. 3). We did not succeed in sequencing a fragment spanning the A + T-rich region and genes flanking the A + T-rich region, and the same as *Orussus occidentalis* and *Cephus cinctus* (Dowton et al., 2009a).

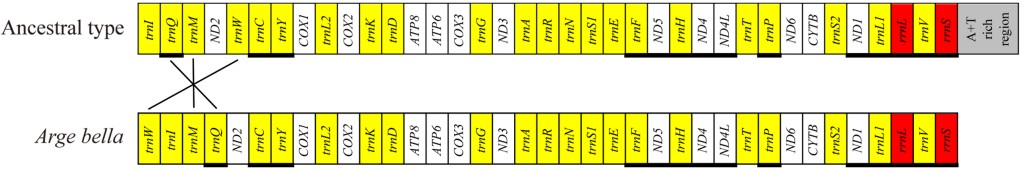

**Figure 3** **Mitochondrial genome organisation of *A. bella*.** Mitochondrial genome organisation of insects in general (above) and *A. bella* (below). Genes are transcribed from left to right, except for those underlined. PCGs are marked by white, the A + T rich region by grey, rRNA genes by red, and tRNA genes by yellow and single-letter amino acid codes. Gene rearrangements are shown with connecting lines that indicate the translocation of *trnW* from the *trnW-trnC-trnY* cluster to a position downstream of trnI and the swapped position of *trnM* and *trnQ*.

There were totally 23 overlapping nucleotides in six locations, and the length of the overlapping sequences ranged from 1 to 7 bp (Table 2). Some overlapping nucleotides were conserved in the *A. bella* mitochondrial genome: ATGATAA between *ATP8* and *ATP6*, and ATGTTAA between *ND4* and *ND4L*, which are also common features of many other insect mitochondrial genomes (*Chai, Du & Zhai, 2012*). There were totally 192 non-coding positions between neighboring genes in 15 locations, and the length of non-coding sequences ranged from 1 to 44 bp (Table 2). There were five locations where the length was over 15 bp: 33 bp between *trnW* and *trnI*, 30 bp between *trnI* and *trnM*, 25 bp between *trnR* and *trnN*, and 44 bp between *COX2* and *trnK*.

## Protein-coding genes and codon usage

All PCGs were initiated by ATN codons: four genes (*ND2*, *ATP8*, *ND3* and *ND6*) used ATA as start codon, four genes (*COX1*, *ND5*, *ND4L* and *ND1*) started with ATT, and five genes (*COX2*, *ATP8*, *COX3*, *ND4* and *CYTB*) were initiated with ATG (Table 2).

The stop codons of *A. bella* were generally TAA, except for *ND6*, which ended with TA, and *COX1*, *COX3* and *ND4*, which ended with T (Table 2). Incomplete stop codons have been reported for all symphytan mitochondrial genomes sequenced to date.

The nucleotide composition of the mitochondrial genome of *A. bella* was A and T rich, with an 80.7% A + T content (Table 3). In PCGs, the highest A + T content was observed in the third codon position, the highest T content in the second codon position and the lowest G content in the third codon position. The highest A + T content was observed in the *ATP8* gene (89.7%).

It has been reported that the parental N strand remains as a single strand for a longer time during replication of mitochondrial genomes, resulting in deamination of A and C (*Reyes et al., 1998*). This leads to an A- and C-skew on the J strand and a T- and G-skew on the N strand. In the case of *A. bella*, we observed that the AT skew was slightly positive (0.0706). On the contrary, GC skew was negative (−0.2708) when considering the whole mitochondrial genome (Table 3), which shows that the occurrence of A was higher than that of T, and the occurrence of C was higher than that of G, which is a general phenomenon in symphytan mitochondrial genomes (*Castro & Dowton, 2005*; *Dowton et al., 2009b*; *Wei et al., 2015*). PCGs on the J strand were slightly T-skewed (−0.0218) and slightly C-skewed

**Table 3  Nucleotide composition of *A. bella* mitochondrial genome.**

| Feature | Length (bp) | A% | C% | G% | T% | A + T% | AT-skew | GC-skew |
|---|---|---|---|---|---|---|---|---|
| Whole genome | 15,576 | 43.2 | 12.2 | 7.0 | 37.5 | 80.7 | 0.0706 | −0.2708 |
| Protein coding genes | 11,222 | 35.6 | 10.7 | 9.9 | 43.8 | 79.4 | −0.1033 | −0.0388 |
| First codon position | 3,741 | 33.2 | 11.7 | 13.0 | 42.1 | 75.3 | −0.1182 | 0.0526 |
| Second codon position | 3,741 | 33.4 | 11.9 | 8.4 | 46.4 | 79.8 | −0.1629 | −0.1724 |
| Third codon position | 3,741 | 40.2 | 8.6 | 8.4 | 42.9 | 83.1 | −0.0325 | −0.0118 |
| Protein coding genes-J | 6,928 | 38.2 | 13.5 | 8.5 | 39.9 | 78.1 | −0.0218 | −0.2273 |
| First codon position | 2,310 | 39.8 | 12.5 | 7.4 | 40.3 | 80.1 | −0.0062 | −0.2563 |
| Second codon position | 2,309 | 39.1 | 14.1 | 9.4 | 37.4 | 76.5 | 0.0222 | −0.2000 |
| Third codon position | 2,309 | 35.7 | 13.8 | 8.6 | 41.9 | 77.6 | −0.0799 | −0.2321 |
| Protein coding genes-N | 4,294 | 31.4 | 6.3 | 12.3 | 50.0 | 81.4 | −0.2285 | 0.3226 |
| First codon position | 1,432 | 34.4 | 3.8 | 12.2 | 49.6 | 84.0 | −0.1810 | 0.5250 |
| Second codon position | 1,431 | 25.6 | 9.4 | 15.2 | 49.8 | 75.4 | −0.3210 | 0.2358 |
| Third codon position | 1,431 | 34.0 | 5.7 | 9.5 | 50.8 | 84.8 | −0.1981 | 0.2500 |
| ATP6 | 675 | 39.1 | 14.4 | 7.0 | 39.6 | 78.7 | −0.0064 | −0.3458 |
| ATP8 | 174 | 40.8 | 9.2 | 1.1 | 48.9 | 89.7 | −0.0903 | −0.7864 |
| COX1 | 1,549 | 34.9 | 14.2 | 12.0 | 38.9 | 73.8 | −0.0542 | −0.0840 |
| COX2 | 681 | 40.7 | 13.7 | 9.0 | 36.7 | 77.4 | 0.0517 | −0.2070 |
| COX3 | 781 | 36.4 | 14.3 | 10.9 | 38.4 | 74.8 | −0.0267 | −0.1349 |
| CYTB | 1,134 | 36.1 | 14.5 | 9.3 | 40.1 | 76.2 | −0.0525 | −0.2185 |
| ND1 | 954 | 49.9 | 14.0 | 6.5 | 29.6 | 79.5 | 0.2553 | −0.3659 |
| ND2 | 1,077 | 42.6 | 11.6 | 4.4 | 41.4 | 84.0 | 0.0143 | −0.4500 |
| ND3 | 354 | 37.6 | 11.9 | 8.2 | 42.4 | 80.0 | −0.0600 | −0.1841 |
| ND4 | 1,339 | 50.0 | 12.7 | 6.0 | 31.3 | 81.3 | 0.2300 | −0.3583 |
| ND4L | 297 | 52.5 | 10.8 | 5.4 | 31.3 | 83.8 | 0.2530 | −0.3333 |
| ND5 | 1,716 | 49.8 | 11.2 | 6.5 | 32.5 | 82.3 | 0.2102 | −0.2655 |
| ND6 | 530 | 41.9 | 12.1 | 5.3 | 40.8 | 82.7 | 0.0133 | −0.3908 |
| rrnL | 1,390 | 45.2 | 11.1 | 5.0 | 38.8 | 84.0 | 0.0762 | −0.3789 |
| rrnS | 839 | 43.7 | 11.6 | 5.4 | 39.3 | 83.0 | 0.0530 | −0.3647 |

(−0.2273), whereas PCGs encoded by the N strand were all slightly T-skewed (−0.2285) and moderately G-skewed (0.3226).

Codon usage in the *A. bella* mitochondrial genome is presented in Table 4. As in other insect mitochondrial genomes (*Foster, Jermiin & Hickey, 1997*), a significant correlation between codon usage and nucleotide composition was found. Leu, Ile, Phe, Met and Ser were most frequently used amino acids (Table 4). UUA-Leu had the highest relative synonymous codon usage (4.83) (Table 4). A relationship between the nucleotide compositions of codon usage and amino acid occurrence was noticed. The relationship can be calculated by the ratio of G + C rich codons (Pro, Ala, Arg, and Gly) and A + T rich codons (Phe, Ile, Met, Tyr, Asn, and Lys). The ratio found in *A. bella* (0.27) is similar to or lower than that of other symphytan species (0.28–0.31) (*Korkmaz et al., 2015*).

**Table 4** Codon usage of 13 PCGs in mitochondrial genome of *A bella*.

| Amino acid | Codon | NO. | RSCU | Amino acid | Codon | NO. | RSCU |
|---|---|---|---|---|---|---|---|
| Phe | UUU | 355 | 1.83 | Tyr | UAU | 147 | 1.76 |
|  | UUC | 32 | 0.17 |  | UAC | 20 | 0.24 |
| Leu | UUA | 459 | 4.85 | End | UAA | 0 | 0 |
|  | UUG | 29 | 0.31 |  | UAG | 0 | 0 |
| Leu | CUU | 31 | 0.33 | His | CAU | 53 | 1.66 |
|  | CUC | 6 | 0.06 |  | CAC | 11 | 0.34 |
|  | CUA | 45 | 0.47 | Gln | CAA | 55 | 1.75 |
|  | CUG | 0 | 0 |  | CAG | 8 | 0.25 |
| Ile | AUU | 396 | 1.81 | Asn | AAU | 214 | 1.75 |
|  | AUC | 42 | 0.19 |  | AAC | 30 | 0.25 |
| Met | AUA | 308 | 1.9 | Lys | AAA | 128 | 1.83 |
|  | AUG | 17 | 0.1 |  | AAG | 12 | 0.17 |
| Val | GUU | 62 | 2.12 | Asp | GAU | 53 | 1.71 |
|  | GUC | 0 | 0 |  | GAC | 9 | 0.29 |
|  | GUA | 52 | 1.78 | Glu | GAA | 66 | 1.74 |
|  | GUG | 3 | 0.1 |  | GAG | 10 | 0.26 |
| Ser | UCU | 87 | 2.21 | Cys | UGU | 39 | 1.9 |
|  | UCC | 7 | 0.18 |  | UGC | 2 | 0.1 |
|  | UCA | 112 | 2.85 | Trp | UGA | 83 | 1.93 |
|  | UCG | 3 | 0.08 |  | UGG | 3 | 0.07 |
| Pro | CCU | 63 | 1.92 | Arg | CGU | 7 | 0.6 |
|  | CCC | 16 | 0.49 |  | CGC | 1 | 0.09 |
|  | CCA | 52 | 1.59 |  | CGA | 36 | 3.06 |
|  | CCG | 0 | 0 |  | CGG | 3 | 0.26 |
| Thr | ACU | 62 | 1.46 | Ser | AGU | 27 | 0.69 |
|  | ACC | 13 | 0.31 |  | AGC | 2 | 0.05 |
|  | ACA | 93 | 2.19 |  | AGA | 75 | 1.91 |
|  | ACG | 2 | 0.05 |  | AGG | 1 | 0.03 |
| Ala | GCU | 55 | 2.08 | Gly | GGU | 32 | 0.7 |
|  | GCC | 8 | 0.3 |  | GGC | 2 | 0.04 |
|  | GCA | 42 | 1.58 |  | GGA | 130 | 2.86 |
|  | GCG | 1 | 0.04 |  | GGG | 18 | 0.4 |

**Notes.**

No, frequency of each codon; RSCU, relative synonymous codon usage.

## Transfer RNA genes

The position and orientation of the predicted tRNAs and anticodon sequences were identical to most of the hitherto reported symphytan mitochondrial genomes (Table 2). 14 tRNAs were encoded by the J strand, and the others by the N strand. All tRNAs folded into a usual clover-leaf structure except for *trnS1* (AGN). Compared with other symphytan species, *trnS1* (AGN) lacked a dihydrouridine (DHU) arm. The size of tRNAs ranged from 64 bp (*trnS1*) to 74 bp (*trnH* and *trnN*) (Fig. 4), placed well within the observed

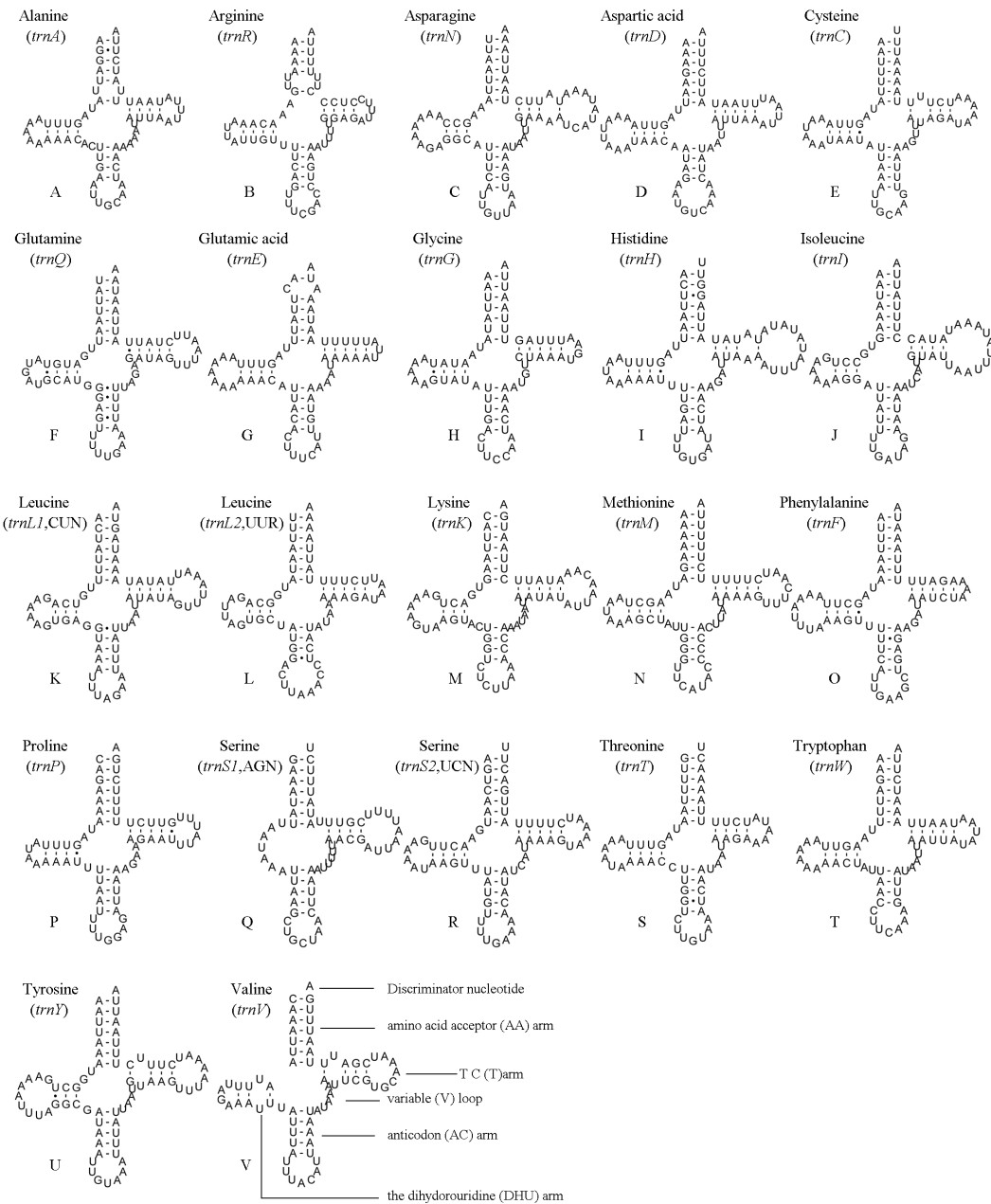

**Figure 4  A. bella tRNAs.**  Predicted secondary structures of the 22 tRNA genes of *A. bella* (A–V): (A) trnA; (B) trnR; (C) trnN; (D) trnD; (E) trnC; (F) trnQ; (G) trnE; (H) trnG; (I) trnH; (J) trnI; (K) trnL1 (CUN); (L) trnL2 (UUR); (M) trnK; (N) trnM; (O) trnF; (P) trnP; (Q) trnS1 (AGN); (R) trnS2 (UCN); (S) trnT; (T) trnW; (U) trnY; (V) trnV. Dashes indicate Watson–Crick base pairing and dots indicate G–U base pairing.

ranges in insects. The observed size differences resulted from changes in the length of the variable loop, dihydrouridine (DHU) arm and T ΨC arm (*Clary & Wolstenholme, 1985*). Anticodon sequences of the tRNA genes were identical with previously-reported symphytan mitogenomes (Table 2). Pairing mismatches occurred mainly in the DHU arm,

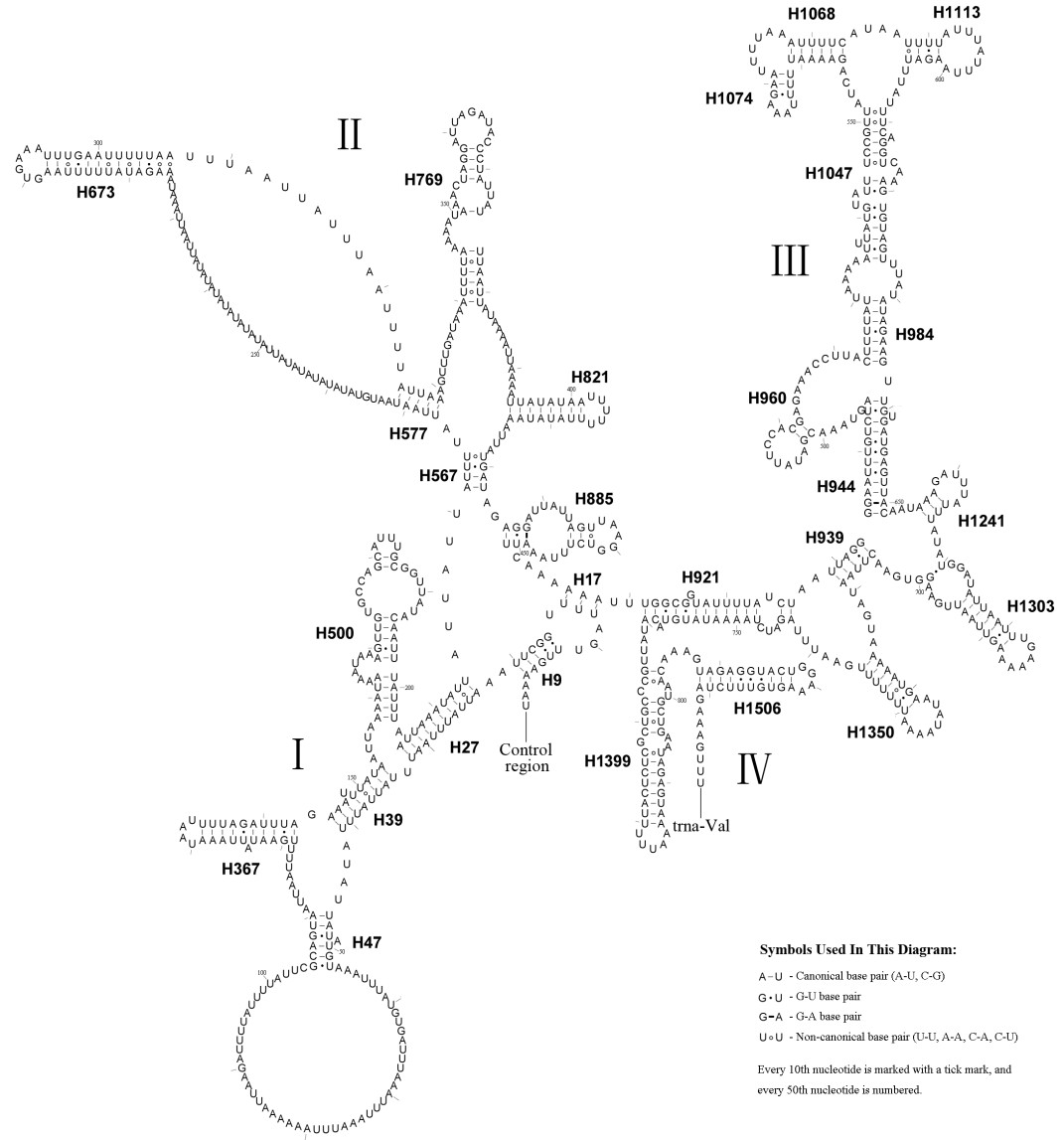

**Figure 5** *A. bella rrnS.* The predicted secondary structure of *rrnS* in the *A. bella* mitochondrial genome. The numbering of helices follows *Gillespie et al. (2006)*, and the domain names follows *Niehuis, Naumann & Misof (2006)*; roman numbers refer to domain names.

AA arm and AC arm, and sometimes in T ΨC arm. All of the 18 mismatches were G-U pairs.

## Ribosomal RNA genes

In *A. bella*, *rrnS* was located downstream of *trnV*, and *rrnL* was located between *trnL1* and *trnV* (Table 2). Both rRNAs were encoded by the N strand, and their lengths were 839 bp and 1,390 bp, and A + T contents 83.0% and 84.0%, respectively.

The *rrnS* secondary structure of *A.bella* contained four domains (*Simon et al., 1994*; *Niehuis, Naumann & Misof, 2006*) and 27 helices (Fig. 5). Previous studies have shown

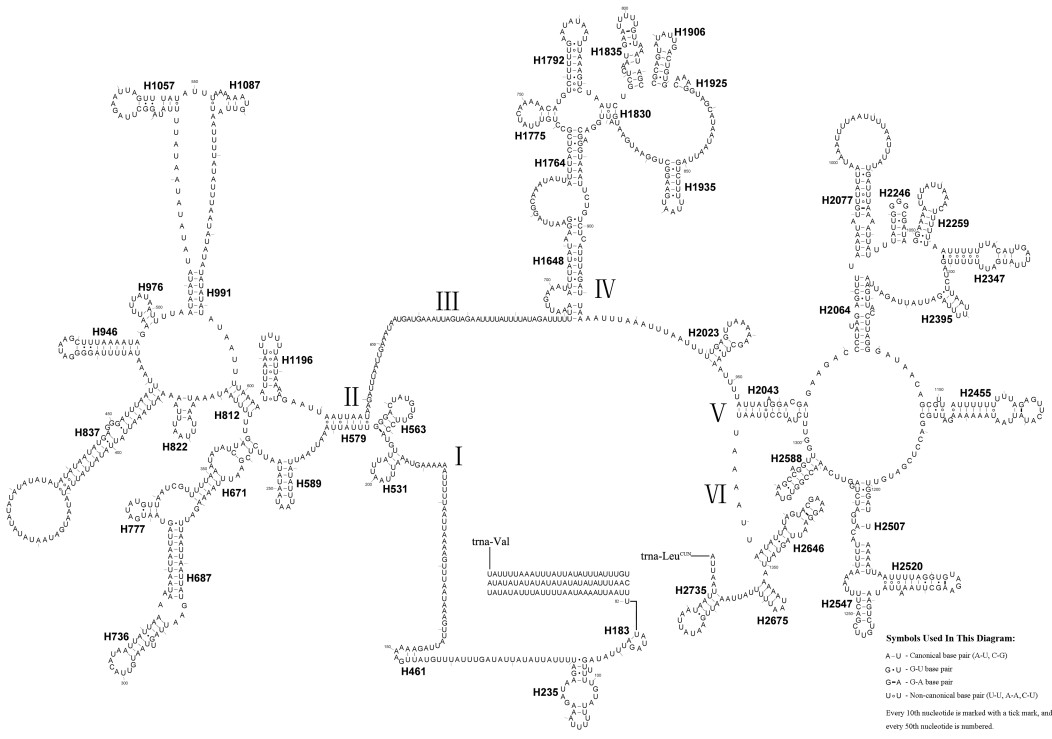

**Figure 6  *A. bella rrnL*.** Predicted secondary structure of *rrnL* in the *A. bella* mitochondrial genome. The numbering of helices and domain names follows *Gillespie et al. (2006)*; roman numbers refer to domain names.

that the second half of the domain III sequence can be difficult to align precisely, even when information on secondary structure is considered. The *rrnS* domain III model of *A.bella* was very similar to the one described in *Hickson et al. (1996)*. H821 was redundant compared with other symphytan species like *Cephus* (*Korkmaz et al., 2015*; *Korkmaz et al., 2016*; *Korkmaz et al., 2017*), which includes 26 helices. The predicted structure of H921 was well conserved in symphytan species, but loop size in H47 is variable. The predicted structures of H500, H769, H944 and H1047 were conserved in symphytan species. H1399 and H1506 helices were well conserved in *A. bella*, as well as in other insect species (*Cameron & Whiting, 2008*; *Cannone et al., 2002*; *Gillespie et al., 2006*; *Misof & Fleck, 2003*; *Wei et al., 2009*; *Wei et al., 2010b*).

The length of the *rrnL* gene was 1,390 bp (Table 1), with an 84.0% A + T content. The secondary structure of the *rrnL* gene in *A. bella* conformed to models proposed for other insects, with the 45 helices belonging to six domains (Fig. 6) (*Cameron & Whiting, 2008*; *Cannone et al., 2002*; *Gillespie et al., 2006*; *Misof & Fleck, 2003*; *Wei et al., 2009*; *Wei et al., 2010a*; *Wei et al., 2010b*). H563, H671, H1925 and H2043 were conserved, and H1775 almost with three pairs in symphytan species. H991 was different from those of *P. condei*, *O. occidentalis*, and *Monocellicampa pruni* with regards to helical length and loop size.

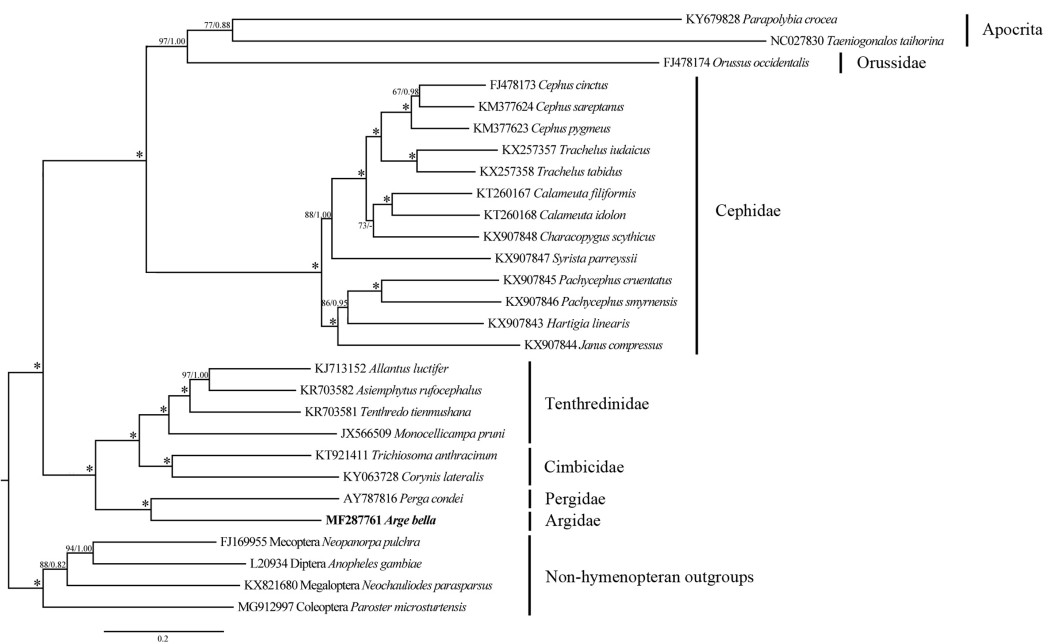

**Figure 7** **Phylogenetic tree of Symphyta and selected apocritan and outgroup taxa, based on a Maximum-likelihood analysis of sequence data from 13 PCGs and 2 rRNA genes.** The numbers at the branches represent Maximum-likelihood bootstrap values/Bayesian posterior probabilities. 100/1.00 is denoted by an asterisk. The scale bar indicates the number of substitutions per site.

## Phylogenetic relationships

We investigated the phylogenetic position of *A. bella* by combining our new mitochondrial genome sequences with previously-reported data from 21 species of Symphyta representing five other families (Table 1), as well as with sequences from two apocritan species and four non-hymenopteran outgroup taxa.

Both ML and BI analyses placed *A. bella* as sister to *Perga condei* with high support (Fig. 7). This Pergidae + Argidae monophylum, as well as its placement as sister to the remaining non-blasticotomid tenthredinoids, which were consistent with the results from comprehensive morphological (*Vilhelmsen, 1997*; *Schmidt et al., 2006*) and molecular (*Malm & Nyman, 2015*) studies. The phylogenetic location of *A. bella* can be considered to constitute the basal branch of the normal Tenthredinoidea (*Ross, 1937*; *Benson, 1938*; *Taeger, Blank & Liston, 2010*), or the suborder Tenthredinomorpha (*Wei & Nie, 1998*). On a wider phylogenetic scale, our results supported a grouping of ((Cephidae, (Orussidae, Apocrita)), ((Argidae, Pergidae), (Cimbicidae, Tenthredinidae))) within the Hymenoptera.

## CONCLUSIONS

*Arge bella* Wei & Du sp. nov. is a new species belonging to *A. vitalisi* group. It is similar to *A. nigricrux Malaise (1943)* and *A. vitalisi Turner (1919)* from south Asia, but differs from

them by the antennal flagellum entirely black, the dorsum of mesonotum mainly blue black, and the abdominal tergites 1, 3–6 each with a large and broad bluish black macula.

The nearly complete mitochondrial genome of *A. bella* (15,576 bp) displays a highly conserved structure and composition as compared to the mitochondrial genomes of other symphytans as well as insects in general. The main differences include minor rearrangements or translocations of three tRNAs, a non-clover-leaf-like structure of *trnS1* (AGN), and redundancy of H821 of *rrnS* and H976 of *rrnL*. ML and BI phylogenetic analyses resulted in a hymenopteran tree with the structure ((Cephidae, (Orussidae, Apocrita)), ((Argidae, Pergidae), (Cimbicidae, Tenthredinidae))) with high nodal supports. Hence, mitochondrial genome sequencing of additional symphytan taxa in the future can clearly produce useful data for resolving hymenopteran relationships.

## ACKNOWLEDGEMENTS

Thanks are due to Ms Wei Songyun of Western Washington University for proofreading of the English. Thanks are due to Dr. Marko Prous of Senckenberg Deutsches Entomologisches Institut for making valuable comments on a previous version of the manuscript. Thanks are also due to Ms Yan Yuchen, Ms Liu Ting, Ms Zhao Hang and Ms Liu Mengmeng of Central South University of Forestry and Technology for collecting the interesting specimens, and to Ms Wu Xiaotong and Ms Zhang Yaoyao for their kind help in this research. Thanks are also due to Ms Tang Min of China Agricultural University for her kind help.

### Funding

This work was supported by the National Natural Science Foundation of China (No. 31501885, No.31672344, and No.30771741). The funders had no role in study design, data collection and analysis, decision to publish, or preparation of the manuscript.

### Grant Disclosures

The following grant information was disclosed by the authors:
National Natural Science Foundation of China: 31501885, 31672344, 30771741.

### Competing Interests

The authors declare there are no competing interests.

### Author Contributions

- Shiyu Du performed the experiments, analyzed the data, prepared figures and/or tables, submit the data to Genbank.
- Gengyun Niu conceived and designed the experiments, analyzed the data, prepared figures and/or tables, approved the final draft, submit the data to Zoobank.
- Tommi Nyman analyzed the data.
- Meicai Wei contributed reagents/materials/analysis tools, authored or reviewed drafts of the paper.

## DNA Deposition

The following information was supplied regarding the deposition of DNA sequences:

The Arge bella sequences are available at GenBank accession number MF287761. The data can also be found in the Supplemental Files and Figshare (https://figshare.com/projects/Characterization_of_the_mitochondrial_genome_of_Arge_bella_sp_nov_Hymenoptera_Argidae_/23515).

## Data Availability

The raw data is available as Supplemental Files and at FigShare:

Du, Shiyu (2018): Partition file of 13 CDS and 2 rRNAs by IQ-TREE. figshare. Dataset.

Du, Shiyu (2018): 13 CDS and 2 rRNA alignment file. figshare. Dataset.

Du, Shiyu (2018): Alignment flie of 13 CDS and 2 rRNAS by IQ-TREE. figshare. Dataset.

Du, Shiyu (2018): Nexus of 13CDS and 2rRNAs by Bayes. figshare. Dataset.

Du, Shiyu (2018): Phylogenetic tree of Symphyta and selected apocritan and outgroup taxa, based on a Maximum-likelihood analysis of sequence data from 13 PCGs and 2 rRNA genes. figshare. Figure.

Du, Shiyu (2018): A Systematic Study of Hartigiinae (Hymenoptera Cephidae) from China. figshare. Paper.

Du, Shiyu (2018): Mitogenome organisation of Arge bella. figshare. Figure.

Du, Shiyu (2018): Arge bella sequence. figshare. Dataset.

Du, Shiyu (2018): Raw data of the specimens (2). figshare. Dataset.

Du, Shiyu (2018): Raw data of the specimens (1). figshare. Dataset.

Du, Shiyu (2018): Arge bella rrnL(16S). figshare. Figure.

Du, Shiyu (2018): Arge bella rrnS (12S). figshare. Figure.

Du, Shiyu (2017): Arge bella tRNAs. figshare. Figure.

Du, Shiyu (2017): Arge bella. figshare. Figure.

Du, Shiyu (2017): Arge bella sp. nov.. figshare. Figure.

https://figshare.com/projects/Characterization_of_the_mitochondrial_genome_of_Arge_bella_sp_nov_Hymenoptera_Argidae_/23515.

## New Species Registration

The following information was supplied regarding the registration of a newly described species:

Publication LSID: urn:lsid:zoobank.org:pub:A94BD62A-D4BE-40F9-8718-84F425875C7C,

Arge bella Wei & Du sp. nov., Genus name: urn:lsid-zoobank.org-act-FFF09ACD-C1BB-440A-90B4-3D56D71FEECD,

Species name: urn:lsid:zoobank.org:act:9AF00C3F-D5EE-474B-BD9E-4794BECACA4F.

## Supplemental Information

Supplemental information for this article can be found online at http://dx.doi.org/10.7717/peerj.6131#supplemental-information.

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
