# Peer review of "Characterization of the mitochondrial genome of Arge bella Wei & Du sp. nov. (Hymenoptera: Argidae)"

_PeerJ, doi:10.7717/peerj.6131_

## Round 0.1 · original submission · Minor Revisions

Dear Dr. Du and colleagues:

Thanks for submitting your manuscript to PeerJ. I have now received two independent reviews of your work, and as you will see, both are very favorable. Well done! Nonetheless, both reviewers raised some relatively minor concerns about the research, and areas where the manuscript can be improved. A few things in particular: 1) make sure missing references are included in your work, 2) make sure that ALL figures and tables contain all of the information necessary to make them “stand alone”, and that your interpretations of these objects are consistent and accurate, and 3) use "unlinked" branch lengths in PartitionFinder analysis and compare with “linked”. Please also remember that reviewer 2 provided a marked-up version of your manuscript.

I agree with the issues raised by the reviewers, and thus feel that their concerns should be adequately addressed before moving forward.

Therefore, I am recommending that you revise your manuscript accordingly, taking into account all of the issues raised by the reviewers. I do believe that your manuscript will be ready for publication once these issues are addressed.

Good luck with your revision,

-joe

·

Basic reporting

Only few corrections/comments in the attached ms.

Experimental design

no comment

Validity of the findings

no comment

·

Basic reporting

This manuscript determined one newly mitochondrial genome of the new species Arge bella, presented comparative analyses of this mitogenome with the other known symphytan mitogenomes and presented a phylogenetic analysis of Symphyta based on the mitogenome sequences. The manuscript is clearly written.

Experimental design

There seems to be no methodological problem.

Validity of the findings

There are important results for the field and the findings are well constructed by authors. The findings have been discussed and supported by relevant literature sufficiently.

Additional comments

Here, the manuscript presents the characterisation of the mitochondrial genome of A. bella for the first time. Then the authors aimed to comparison of various genomic features against other reported symphytan species. The most interesting evidence is the presence of rearrangements in tRNA genes. The manuscript presents a newly constructed phylogenetic tree using available symphytan mitochondrial genome to validate the phylogenetic position of the species (A. bella). Some minor changes or comments are listed below:
• Line 60: The order of references should be updated. For example, Dowton et al. 2009a should be used at first referenced, not Dowton et al., 2009b. This problem should be checked throughout the manuscript.
• Lines 62-63: I could not see the family name Heptameliidae in ECatSym database as well as the current literatures.
• Line 164: The authors should use "unlinked" branch lengths in the analysis with PartitionFinder. When we prefer "linked" branch lengths, we're assuming that the relative rates of evolution among lineages in the tree are constant across partitions. However, with "unlinked" branch lengths, not only can the partitions have different overall rates of evolution, but also the relative rates of evolution among lineages can differ between partitions.
• Lines 247-249: The authors should explain the likely reasons of this failure together with the relevant literature.
• Line 263: When looking Table 2, Leu, Ile, Phe, Met, Ser seem to be most frequently used amino acids rather than indicated by the authors.
• Line 304: When examining the related figure, it seems to have three domains. Was it wrongly denoted, or not, please give more explanation about this new domain. As taking into consideration the relevant literature, the predicted secondary structure of insect rrnS gene seems to have three domains.
The remaining minor corrections are shown on the manuscript.
In my opinion, the manuscript could be published in the journal PeerJ after minor revision.

Reviewer 3 ·

Basic reporting

Du et al. have sequenced the mitochondrial genome of Arge bella and compare with others.
Totally the paper has been analysed and written in good order. it is including new information of mitochondrial genome of Arge bella. The hypotheses of paper is unclear and it need to explain clearly in the last paragraph of INTRODUCTION, especially when the paper is a comparative study (Just the text needs to revise)

Experimental design

It is an Original primary research and the METHODS AND MATERIALS have been clearly explained and designed.

Validity of the findings

This paper give new information about mitochondrial genome of Arge bella.

Additional comments

There is a major scientific question that should be reconsider by authors. In the title, Arge bella was mentioned as new species (sp. nov.). It means this is the first time that this name used and author discussed about its taxonomical status for first time. But in page 4 line 179, this species was referred to "Arge bella Wei & Du sp. nov." if Wei & Du have described this species before this paper so "sp. nov." should be removed from title otherwise line 179 should be revised.

in Fig. 7: Generally values of Bayesian posterior probabilities were mentioned in the range of 0 to 1. I suggest authors follow in this way.

In Tab 3. I suggest the column of Anticodon replace with Codon
In Tab 3. I suggest J and M replace with H (Heavy) and L (Light)

---

## Round 0.2 · accepted · Accept

Dear Dr. Du and colleagues:

Thanks for re-submitting your manuscript to PeerJ, and for addressing the concerns raised by the reviewers. I now believe that your manuscript is suitable for publication. Congratulations! I look forward to seeing this work in print, and I anticipate it being an important resource for hymenopterists with interests in mitochondrial genomics. Thanks again for choosing PeerJ to publish such important work.

-joe

#